# Subjective and psychophysical olfactory and gustatory dysfunction among COVID-19 outpatients; short- and long-term results

Mads Mose Jensen[1]*, Kasper Daugaard Larsen[1,2], Anne-Sophie Homøe[1,3], Anders Lykkemark Simonsen[4], Elisabeth Arndal[2], Anders Koch[4,5], Grethe Badsberg Samuelsen[3], Xiaohui Chen Nielsen[6], Tobias Todsen[1,2,7], Preben Homøe[1,7]

1 Department of Otorhinolaryngology and Maxillofacial Surgery, Zealand University Hospital, Koege, Denmark, 2 Department of Otolaryngology—Head and Neck Surgery and Audiology, Rigshospitalet, Copenhagen, Denmark, 3 Department of Otorhinolaryngology, Nordsjaellands Hospital, Hilleroed, Denmark, 4 Department of Infectious Diseases, Rigshospitalet, Copenhagen, Denmark, 5 Staten Serum Institut (SSI), Copenhagen, Denmark, 6 Department of Clinical Microbiology, Zealand University Hospital, Koege, Denmark, 7 Department of Clinical Medicine, University of Copenhagen, Copenhagen, Denmark

* madsmose@gmail.com

**Data Availability Statement:** Due to our nation's strict data sharing policies, we are unable to make our data public, even in pseudo anonymized form. However, researchers who wish to access the data

## Abstract

### Background

Olfactory and gustatory dysfunctions are early symptoms of SARS-CoV-2 infection. Between 20–80% of infected individuals report subjective altered sense of smell and/or taste during infection. Up to 2/3 of previously infected experience persistent olfactory and/or gustatory dysfunction after 6 months. The aim of this study was to examine subjective and psychophysical olfactory and gustatory function in non-hospitalized individuals with acute COVID-19 up to 6 months after infection.

### Methods

Individuals aged 18-80-years with a positive SARS-CoV-2 PCR test no older than 10 days, were eligible. Only individuals able to visit the outpatient examination facilities were included. Gustatory function was tested with the Burgharts Taste Strips and olfactory function was examined with the Brief Smell Identifications test (Danish version). Subjective symptoms were examined through an online questionnaire at inclusion, day 30, 90 and 180 after inclusion.

### Results

Fifty-eight SARS-CoV-2 positive and 56 negative controls were included. 58.6% (34/58) of SARS-CoV-2 positive individuals vs. 8.9% (5/56) of negative controls reported subjective olfactory dysfunction at inclusion. For gustatory dysfunction, 46.5% (27/58) of positive individuals reported impairment compared to 8.9% (5/56) of negative controls. In psychophysical tests, 75.9% (46/58) had olfactory dysfunction and 43.1% (25/58) had gustatory dysfunction among the SARS-CoV-2 positive individuals at inclusion. Compared to negative controls, SARS-CoV-2 infected had significantly reduced olfaction and gustation. Previously

can do so by contacting Professor Preben Homøe (prho@regionsjaelland.dk) at the Department of Otorhinolaryngology Zealand University Hospital or the Regional Data Protection Agency (forskningfortegnelse@regionsjaelland.dk) to make a data sharing contract. After permission has been given by the regional data committee, data will be made available to the researchers.

**Funding:** Yes - PH has received funding from The EU Interreg ØKS foundation (https://interreg-oks. eu/) grant number NYPS 20303399. The sponsor had no role in planning, execution or analyzing the results of the study.

**Competing interests:** The authors have declared that no competing interests exist.

infected individuals continued to report lower subjective sense of smell 30 days after inclusion, whereafter the difference between the groups diminished. However, after 180 days, 20.7% (12/58) positive individuals still reported reduced sense of smell and taste.

## Conclusion

Olfactory and gustatory dysfunctions are prevalent symptoms of SARS-CoV-2 infection, but there is inconsistency between subjective reporting and psychophysical test assessment of especially olfaction. Most individuals regain normal function after 30 days, but approximately 20% report persistent olfactory and gustatory dysfunction 6 months after infection.

## Background

Shortly after the emergence of the novel coronavirus, SARS-CoV-2, olfactory (OD) and gustatory dysfunction (GD) proved to be early symptoms of Coronavirus Disease 2019 (COVID-19) [1]. The prevalence of OD ranges considerably from 5–80% of infected [2, 3]. However two separate meta-analysis show a similar prevalence at around 50% [4, 5]. Several studies have confirmed the relationship between OD and COVID-19, and sudden onset OD was even suggested as a predictor for COVID-19 in otherwise asymptomatic individuals [6, 7]. While OD has proved a frequent symptom of COVID-19, less focus has been put on GD. There have been speculations that GD was merely a side effect to OD, while other reports show a high prevalence and a distinct pathogenesis [8–10]. The pathogenesis for OD in COVID-19 is unknown, but it differs from other respiratory infections by being independent of nasal congestion [11, 12]. Instead, local inflammation caused primarily by the binding of SARS-CoV-2 virus to the ACE-2 and TMPRSS2 receptors in the sustentacular (non-neural) cells of the olfactory epithelium seems crucial [13, 14]. Furthermore, a more aggressive immune response can cause damage to the neuronal cells, prolonging the period of olfactory dysfunction with a risk of permanent anosmia [13]. Recent radiological finding also suggest that post-COVID-19 OD can possibly lead to olfactory bulb atrophy [13, 15]. The duration of OD differs greatly among COVID-19 infected individuals. Many previously infected regain normal olfactory function after a few weeks [16, 17], and some studies report complete recovery after 6 months [16, 18]. However, several other studies report persistent hyposmia 6 months after primary infection in 20–75% [17, 19–22], and up to a quarter of previously infected still experience olfactory dysfunction one year following infection [23–25].

When assessing olfactory and gustatory function, it is important to recognize that subjective reporting of olfactory and gustatory dysfunction generally corresponds poorly to psychophysical test measurements [26–28].

This study aims to compare the degree of subjective and psychophysical GD and OD in individuals with or without SARS-CoV-2 infection from June 2020 –May 2021 and to monitor subjective recovery up to 6 months after infection.

## Methods

This study was a longitudinal prospective case-control study of OD and GD in SARS-CoV-2 positive individuals and negative controls.

## Subjects

Participants were recruited from the Capital Region and Region Zealand in Denmark from June 2020 to May 2021. Adults 18-80-years old with a positive SARS-CoV-2 Polymerase Chain Reaction (PCR) test less than 10 days old, were included either via information posters at test facilities in the Capital Region and Region Zealand, or via direct contact through the Danish governmental online communication platform 'E-boks'. Both ways of inclusion led participants to an online REDCap (Research Electronic Data Capture hosted at the Capital Region, Denmark) questionnaire with further information about the study and a simple yes/no question regarding recent loss of smell and taste. Participants were then contacted by a doctor and invited for psychophysical olfactory and gustatory assessment at either Zealand University Hospital, Koege, Nordsjaellands Hospital, Hilleroed or Rigshospitalet, Copenhagen. Only individuals able to visit the outpatient examination facilities were included. Age-matched negative controls were included through test facilities in both the Capital Region and Region Zealand. Participants who could not fully comprehend the questionnaire and examination was excluded.

The study was conducted in accordance with the Helsinki II Declaration. Written and verbal informed consent was obtained from all participants and the protocol was approved by the Danish Regional Ethics Committee *(protocol number*: *SJ-714* and *Letter ID 4336732)*.

## Psychophysical tests

SARS-CoV-2 positive individuals were examined in special facilities designed to accommodate the risk of infection transmission. Negative controls were examined at the otorhinolaryngological outpatient clinics of Zealand University Hospital, Koege, Nordsjaellands Hospital, Hilleroed or Rigshospitalet, Copenhagen. Enrollment was performed maximum 10 days after the initial SARS-CoV-2 PCR test. A trained otorhinolaryngologist assessed the nasal and oral cavity and noted any anatomical variations that could influence the sense of smell or taste. Three further swabs from the nasal cavity, the rhinopharynx and the oropharynx respectively, were performed on both cases and controls and sent for SARS-CoV-2 PCR analysis at the department for Clinical Microbiology at Zealand University Hospital to confirm SARS-CoV-2 positivity or negativity. Information regarding previous infections with SARS-CoV-2 was not available.

We tested gustation using the Burgharts Taste Strips (Burghart Messtechnik GmbH), containing the basic flavors: sweet, sour, bitter and salty. Failure to recognize all 4 flavors was considered as gustatory dysfunction. For olfactory assessment, participants completed the Brief Smell Identification Test™ (Sensonics International Cross-Cultural Smell ID Test, version A (validated in Danish) (BSIT A)). The degree of psychophysical OD (BSIT test) was categorized into either normosmia (BSIT score ≥9/12), hyposmia (≤8/12 - ≥6/12) or anosmia (≤5/12).

Psychophysical testing of taste and smell was done only once per participant at enrolment.

## Subjective olfactory and gustatory function

Prior to inclusion, participants answered a simple yes/no question regarding olfactory and gustatory dysfunction in the 7 days preceding their answer. At inclusion participants answered an online REDCap questionnaire including general demographic information, previous or current history of OD or GD, if they could smell sweat, perfume and coffee and if they could taste coffee, salt, sweet, bitter and sour (Y/N). Furthermore, participants were asked about any predisposing illnesses or conditions such as, allergy, chronic rhinosinusitis, congested nose and tobacco use. The questionnaire contained additional questions from a validated smell test questionnaire and the nasal symptoms related questions 1–6 from SNOT-22 [29]. Finally,

participants were asked to rank their sense of smell and taste on a numeric rank scale (NRS) with scores from 0–10 (10 being normal). Invitations to the online questionnaire was sent via email at the time of inclusion and at days 30, 90 and 180 following inclusion. For non-responders, up to two-reminder emails were sent.

**Statistical methods.** Median scores for both psychophysical and subjective measurements of OD and GD were calculated. For test of difference of smell and taste test results in cases and controls Pearson chi squared test and Mann-Whitney test were used for categorical and continuous variables, respectively. P-values <0.05 were considered significant. Results for both OD and GD were stratified according to gender. Sensitivity and specificity for psychophysical OD as a predictor for SARS-CoV-2 infections were calculated along with positive predictive value. We used Stata 13 (StataCorp. 2013. Stata Statistical Software: Release 13. College Station, TX: StataCorp LP.).

## Results

In total, 58 SARS-CoV-2 PCR positive individuals and 56 negative controls were included. There was no difference in age and gender distribution amongst the two groups (Table 1). Amongst the SARS-CoV-2 positive, 41.4% (24/58) reported nasal congestion and 34.5% (20/58) reported increased nasal drip while only 16.1% (9/56) of negative controls reported congestion and 14.3% (8/56) reported increased nasal drip. The number of daily tobacco users (1 among cases and 3 among controls) was too low to make stratification. Table 1 also shows other factors that could influence OD/GD.

### Olfactory dysfunction

**Psychophysical results.** BSIT results showed marked differences in smell between SARS-CoV-2 positives and negatives with 55.2% (32/58) and 1.8% (1/56) being anosmic in the two

**Table 1. Demographic characteristics and risk factors.**

|  | SARS-CoV-2 positive | SARS-CoV-2 negative |
| --- | --- | --- |
| Number | 58 | 56 |
| Male n (%) | 30 (51.7) | 29 (51.8) |
| Mean age (Years) | 43.3 | 43.9 |
| Reported symptoms and risk factors of OD/GD | | |
| Previous reduced olfaction or gustation n/N (%) | 2/58 (3.5) | 1/56 (1.8) |
| Daily tobacco use n/N (%) | 1/58 (1.7) | 3/56 (5.4) |
| Increased nasal drip during the last 7 days n/N (%) | 20/58 (34.5) | 8/56 (14.3) |
| Nasal congestion during the last 7 days n/N (%) | 24/58 (41.4) | 9/56 (16.1) |
| Chronic sinusitis or nasal polyps n/N (%) | 2/58 (3.5) | 3/56 (5.4) |
| Hay fever during the last 7 days n/N (%) | 3/58 (5.2) | 4/56 (7.1) |
| Daily use of nasal steroid spray n/N (%) | 2/58 (3.5) | 4/56 (7.1) |
| Daily use of "over-the-counter" decongestant nasal spray n/N (%) | 2/58 (3.5) | 4/56 (7.1) |
| ENT assessment | | |
| Septal deviation n/N (%) | 3/58 (5.2) | 2/56 (3.6) |
| Inflamed nasal mucosa n/N (%) | 4/58 (6.9) | 2/56 (3.6) |

Demographic information and risk factors for reduced sense of smell and taste for SARS-CoV-2 Polymerase Chain Reaction positive cases and negative controls. OD: Olfactory dysfunction, GD: Gustatory dysfunction, ENT: Ear Nose Throat

**Table 2. Subjective loss of smell and taste and psychophysical olfactory dysfunction and gustatory dysfunction assessment.**

| | SARS-CoV-2 positive | SARS-CoV-2 negative | P-value |
|---|---|---|---|
| Total number | 58 | 56 | |
| Subjective loss of smell at inclusion n (%) | 34 (58.6) | 5 (8.9) | <0.001 |
| Subjective loss of taste at inclusion n (%) | 27 (46.5) | 5 (8.9) | <0.001 |
| Brief Smell Identification Test (BSIT) | | | |
| Anosmia (BSIT 0–5 correct) n (%) | 32 (55.2) | 1 (1.8) | <0.001 |
| Hyposmia (BSIT 6–8 correct) n (%) | 12 (20.7) | 11 (19.6) | |
| Normosmia (BSIT 9–12 correct) n (%) | 14 (24.1) | 44 (78.6) | |
| Median BSIT score (25–75% IQR) | 4.5 (1–8) | 10 (9–11) | <0.001 |
| Median BSIT score, females (25–75% IQR) | 4.5 (2–9.5) | 10 (9–11) | <0.001 |
| Median BSIT score, males (25–75% IQR) | 4.5 (1–8) | 10 (9–11) | <0.001 |
| Median BSIT score for subjective anosmia/hyposmia (25–75% IQR) | 4 (1–8) | 8 (7–10) | 0.038 |
| Median BSIT score for subjective normosmia (25–75% IQR) | 7 (3–10) | 10 (9–11) | 0.002 |
| Burgharts taste strips | | | |
| Correctly identify 4/4 taste strips n (%) | 33/58 (56.9) | 42/56 (75) | <0.001 |
| Median taste test (IQR) | 4 (3–4) | 4 (3.5–4) | 0.03 |
| Median taste test, females (IQR) | 4 (3–4) | 4 (4–4) | 0.003 |
| Mean teste test, males (IQR) | 4 (2–4) | 4 (3–4) | 0.57 |

Subjective loss of smell and taste answered before inclusion and psychophysical olfactory and gustatory assessment. Median scores with Interquartile Ranges (IQR) for the Brief Smell Identification Test (BSIT) and Burgharts taste strips for SARS-CoV-2 positive and negative controls stratified for gender.

groups, respectively (Table 2). In total, 75.9% (44/58) of SARS-CoV-2 positives had either anosmia or hyposmia. Median BSIT score was significantly lower for SARS-CoV-2 positive compared to negative and did not differ between sex (Table 2). Table 3 shows the test performance of BSIT as a diagnostic test for SARS-CoV-2.

**Subjective results.** The rate of self-reported hyposmia/anosmia among SARS-CoV-2 positives and negative controls who answered the questionnaire can be seen in Table 4. At the initial questionnaire before inclusion, 58.6% (34/58) of SARS-CoV-2 positive individuals reported subjective OD, compared to 8.9% (5/56) of negative controls (Table 2). Six months after primary infection 20.7% (12/58) reported persistent hyposmia/anosmia (Table 4). The test performance of subjective hyposmia/anosmia as a diagnostic test to detect SARS-CoV-2 can be seen in Table 3.

The subjective OD score (NRS from 0–10, 0 being anosmia) are depicted in Fig 1. The median subjective OD score for SARS-CoV-2 positive was lower than median scores for negative controls at inclusion and day 30. At day 90 and 180, median scores for cases and controls showed no difference (Table 4).

**Table 3. Subjective and psychophysical test performance.**

| | Subjective olfactory dysfunction | Psychophysical olfactory dysfunction |
|---|---|---|
| Sensitivity | 75.9% | 58.6% |
| Specificity | 78.6% | 91.1% |
| Positive Predictive value | 78.5% | 87% |

Sensitivity, specificity and positive predictive values of subjective and psychosocial olfactory dysfunction as diagnostic test for SARS-CoV-2 positivity.

**Table 4. Subjective smell and taste reports and numerical rank scale (NRS) score at 0, 30, 90 and 180 days after follow-up.**

|  | SARS-CoV-2 Positive | SARS-CoV-2 Negative |
|---|---|---|
| Answer rate | | |
| 0 days | 51/58 (87.9) | 44/56 (78.6) |
| 30 days | 39/58 (67.2) | 31/56 (55.4) |
| 90 days | 19/58 (32.8) | 25/56 (44.6) |
| 180 days | 36/58 (62.1) | 20/56 (35.7) |
| Number and (%) reporting subjective anosmia or hyposmia | | |
| 0 days | 34/51 (66.7) | 5/44 (11.4) |
| 30 days | 27/38 (71.1) | 2/31 (6.5) |
| 90 days | 9/19 (47.4) | 1/25 (4.0) |
| 180 days | 12/36 (33.3) | 1/20 (5.0) |
| Median self-reported smell function from NRS scale 0–10 (IQR) | | |
| 0 days | 3 (1–8) | 10 (9.5–10) |
| 30 days | 8.5 (7–10) | 10 (10–10) |
| 90 days | 10 (7–10) | 10 (10–10) |
| 180 days | 10 (8–10) | 10 (9.5–10) |
| Median self-reported smell function for people with subjective anosmia or hyposmia from NRS scale 0–10 (IQR) | | |
| 0 days | 2,5 (1–5) | 3 (1–8) |
| 30 days | 8 (6–9) | 7.5 (5–10) |
| 90 days | 7 (6–9) | 8 (8–8) |
| 180 days | 7,5 (6–9.5) | 7 (7–7) |
| Number and (%) reporting subjective gustatory dysfunction | | |
| 0 days | 27/50 (54.0) | 5/44 (11.4) |
| 30 days | 20/39 (51.3) | 2/30 (6.7) |
| 90 days | 8/19 (42.1) | 1/25 (4.0) |
| 180 days | 12/36 (33.3) | 1/20 (5.0) |
| Median self-reported taste from NRS scale 0–10 (IQR) | | |
| 0 days | 7 (3–10) | 10 (9–10) |
| 30 days | 9 (8–10) | 10 (10–10) |
| 90 days | 10 (8–10) | 10 (10–10) |
| 180 days | 10 (9–10) | 10 (9.5–10) |
| Median self-reported taste for people with subjective gustatory dysfunction from NRS scale 0–10 (IQR) | | |
| 0 days | 3 (2–7) | 4 (1–8) |
| 30 days | 8.5 (6.5–10) | 7.5 (5–10) |
| 90 days | 8 (8–10) | 8 (8–8) |
| 180 days | 8.5 (8–10) | 7 (7–7) |

Questionnaire answer rates and subjectively reported smell and taste impairment at inclusion, day 30, 90 and 180 for SARS-CoV-2 positive and negative controls including median smell and taste scores and Inter Quartile Range (IQR), calculated from subjectively reported smell and taste function ranked in a Numeric Rank Scale score (10 being normal). Median results also stratified for subjective olfactory and gustatory D.

## Gustatory dysfunction

**Objective results.** In total, 75% (42/56) of negative controls recognized 4/4 taste strips correctly, while only 56.9% (33/58) of SARS-CoV-2 correctly identified all taste strips (Table 2). Median Burkharts taste strip scores were 4 in both SARS-CoV-2 positives and negative controls.

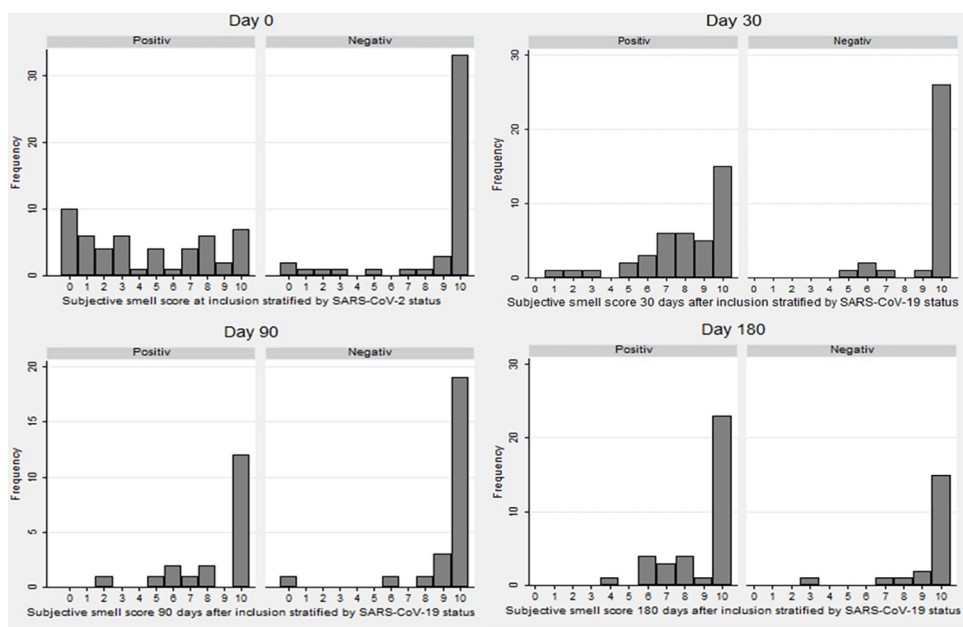

**Fig 1. Subjective smell scores at inclusion, day 30, 90 and 180.** Subjective smell score at inclusion, 30, 90 and 180 days after inclusion, stratified by SARS-CoV-2 status. Numeric Rank Scale where 0 is anosmia and 10 is normal sense of smell.

**Subjective results.** At the initial questionnaire before inclusion, 46.5% (27/58) of SARS-CoV-2 positive participants reported subjective GD compared to 8.9% (5/56) of negative controls. Median subjective NRS taste scores for SARS-CoV-2 positives and negative controls can be seen in Table 4 and Fig 2. Six months after primary infection 20.7% (12/58) reported persistent GD.

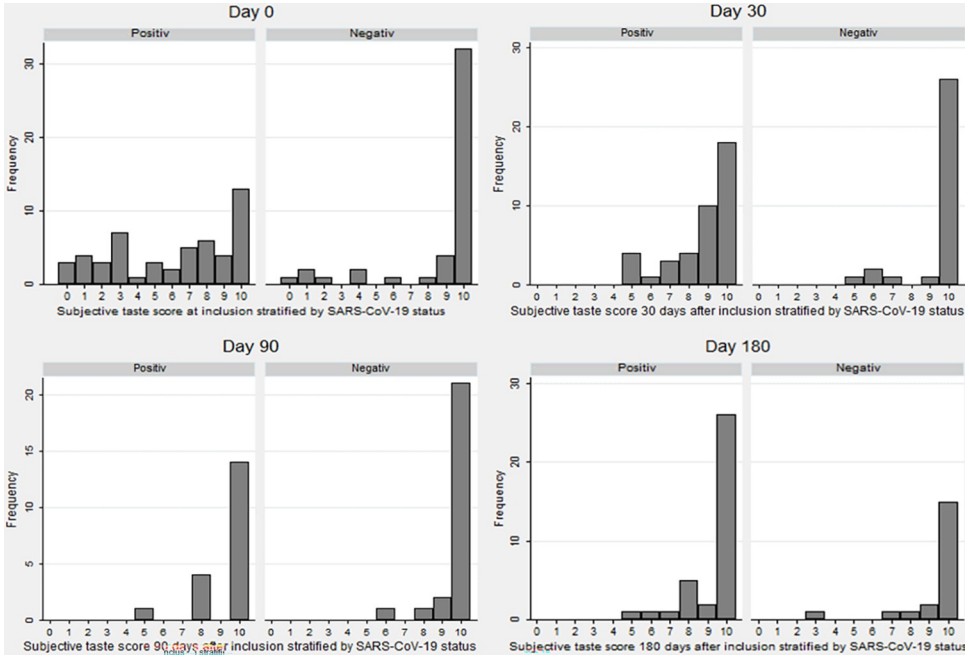

**Fig 2. Subjective taste scores at inclusion, day 30, 90 and 180.** Subjective taste score at inclusion, 30, 90 and 180 after inclusion, stratified by SARS-CoV-2 status. Numeric Rank Scale where 0 is anosmia and 10 is normal sense of taste.

## Discussion

As previously reported [1, 2], the prevalence of OD and GD among SARS-CoV-2 positive individuals varies considerably. In this study, 58.6% of SARS-CoV-2 positive participants reported subjective OD before inclusion, but a higher fraction (75.9%) had either hyposmia or anosmia in objective BSIT tests, which is similar to results from other studies [2]. Likewise, SARS-CoV-2 positive participants had a significantly lower BSIT median score (4.5) than SARS-CoV-2 negatives. Anosmia was seen amongst 55.2% of SARS-CoV-2 positive participants at inclusion, compared to only 1.8% of negative controls. Compared to other studies [30] the median score for BSIT was slightly lower for our control group.

SARS-CoV-2 positive individuals who reported subjective OD had significantly lower BSIT values. This is not surprising, as objective loss of smell could be expected in persons indicating they had lost their sense of smell. However, also SARS-CoV-2 positive individuals who reported normal olfactory function, had loss of smell when measured objectively, suggesting that some individuals fail to recognize their OD.

As previously stated [1], OD has been suggested as an early indicator of SARS-CoV-2 infection. Our results support previous findings, and indicate that subjectively reported OD and GD are less reliable than objective testing as markers of SARS-CoV-2 infection, as 41.4% of SARS-CoV-2 positive individuals did not have subjective loss of smell. This discrepancy is also known from the general population, where psychophysical tests reveal up to 20% of the background population suffer from olfactory dysfunction, but only around 10% report subjective olfactory dysfunction, suggesting many individuals do not have a clear perception of their sense of smell [31]. The large discrepancy between subjective vs. psychophysical assessments underlines the importance of using psychophysical/objective measurements if OD should be used for early SARS-CoV-2 diagnosis. This is also supported by the fact that psychophysical assessment had higher sensitivity (75.9%) compared to subjective measurement (58.6%). However, the specificity for psychophysical assessment was lower compared to subjective assessment. The Positive Predictive value of BSIT as a diagnostic tool for SARS-CoV-2 positives was slightly higher compared to subjective OD. Thus, an individual reporting subjective sudden onset OD would therefore likely be infected with SARS-CoV-2, but the low sensitivity of subjective reporting would mean a large proportion would be overlooked. However, if OD should be used as a predictive marker for SARS-CoV-2, subjective assessment would likely be the only logistically viable solution due to the large amount of patients. It is, however, crucial to recognize that the absence of OD is a poor marker for SARS-CoV-2 negativity.

During follow-up, OD was reported by 71.1% at day 30, and decreased to 33.3% at day 180 for previous SARS-CoV-2 infected, compared to 6.5% at day 30 and 5.0% at day 180 for negative controls. However, the median subjective NRS score for SARS-CoV-2 positives increased from inclusion to day 30. For negative controls, the subjective NRS score remained stable throughout the follow up period. This shows that even though 20.7% of previously infected continue to report OD, the degree of OD was reduced after 30 days. This is consistent with other results [16–22, 32] where OD is seen to reduce after 30 days in previously infected subjects, while up to a quarter of previously infected report persistent olfactory dysfunction. For the group who reported persistent OD, the median score also increased after 30 days, suggesting some degree of improvement despite persistent subjective OD. This can also be seen in Figs 1 and 2, where the amount of SARS-CoV-2 positive individuals reporting low scores are higher at inclusion and day 30 but normalizes at day 90 and 180. While there is still some uncertainty how the long term recovery differs between different strains of SARS-CoV-2 and how vaccinations influences the long term recovery [33], the high rates of persistent olfactory dysfunction is very concerning. A large proportion of the world's populations is likely

previously infected with SARS-CoV-2, and the total number of individuals at risk of persistent chemosensetive dysfunction is enormous. As these individuals face reduces quality of life, and there is no international consensus of treatment, we face a significant public concern [34].

There were no differences between cases and controls regarding nasal septal deviations or self-reported chronic sinusitis, hay fever, daily use of steroid or local decongestant nasal spray. The rate of daily tobacco users was low in both cases and controls. While tobacco use is known to severely reduce olfaction, the number of users was too low to make stratification. SARS-CoV-2 positive individuals had increased levels of nasal congestion and nasal drip, both of which would increase the level of OD/GD due to obstruction of airflow (Table 1). However, the rates of SARS-CoV-2 positive individuals with OD/GD are much higher than the rates of nasal congestion and nasal drip. This supports that COVID-19 related OD is not solely due to blocked nasal flow.

The gold standard for testing olfactory function in the Western World is the Threshold, Discrimination and Identification (TDI) test and The University of Pennsylvania Smell Identification Test (UPSIT). Compared to these tests, the BSIT has reduced sensitivity but is a cost and time-effective method for screening of large groups of patients [30, 35]. Furthermore, due to the contamination risk when testing SARS-CoV-2 positive individuals, reusing test material is problematical, and we find the single use BSIT is preferable. Likewise, threshold taste strips is preferred [36, 37], but the simple 4 flavor taste strips with equal high concentration is a viable solution when screening larger groups.

In our study, participants with a self-reported history of OD/GD was not excluded. Likewise, smokers and individuals with deviated septum were not excluded. While we recognize this means that we cannot conclude SARS-CoV-2 is the cause of OD/GD for people with prior history or risk factors for OD/GD, we partly account for this by having a comparable control group. Furthermore, all statistical analysis were performed where all individuals with prior OD/GD, septal deviation, smoking history was excluded. This did not significantly change the results or conclusions and it was decided to keep them in the final analysis.

## Limitations

During inclusion, answers for BSIT, Burgharts taste strips and objective clinical assessment were noted onto an answer sheet. Unfortunately, the correct answer for the 12[th] BSIT question was not correctly noted on the answer sheet. This meant a case or control who recognized the correct scent would not receive the correct point. We examined this fault by excluding the 12[th] answer in the statistical analysis. This did not change any of the conclusions of the study, and therefore the 12[th] answer was kept in the final results.

It is important to recognize that BSIT and other psychophysical tests are not true objective test of an individual's olfaction. However, true objective test for olfactory dysfunctions, such as olfactory evoked potential, is not ideal for large group of patients and especially not for SARS-CoV-2 infected individuals. Therefore psychophysical testing is often used as a tool for quantifying sense of smell. This quantification is very useful, and often uncovers olfactory dysfunctions not recognized by the patient. However, when interpreting the results one should recognize that answers depend on the individual performing the test.

Our follow up was limited to subjective answers from questionnaires. As already describes, there is a discrepancy between subjective and psychosocial testing, and the long term recovery results should be interpreted with care. The answer rates of the follow-up questionnaires were generally higher for previously SARS-CoV-2 infected individuals. Individuals who experienced OD or GD might have been more inclined to participate and answer the questionnaire, which could possibly lead to inclusion bias. Likewise, participants with persistent OD of GD could be

more inclined to answer follow-up questionnaires. This could overestimate the proportion with OD or GD. Furthermore, we have no information about possible SARS-CoV-2 infection or re-infection in the follow-up period.

Recent anecdotal reports from the latest omicron variant outbreak suggest OD/GD might not be as prevalent as with the original Wuhan virus strain. However, as our inclusion period ran one year from June 2020 to May 2021, at least four different strains were recorded in Denmark (Wuhan, Alpha, Beta, and Gamma). However, mass sequencing was not performed at this time, and therefore we have no information regarding the SARS-CoV-2 strains among the infected participants.

## Conclusion

In this case-control study of outpatient SARS-CoV-2 positive individuals without severe COVID-19 disease, 75.9% had objective olfactory and gustatory dysfunction in the acute phase of the infection. Slightly less reported acute subjective olfactory (58.6%) and gustatory (46.5%) dysfunction, but this remained over time with 20% still reporting subjective OD and GD six months after primary infection. Our study contributes significantly to the knowledge of OD and GD dysfunction during active infection and up to 6 months after. We tested both objective and subjective OD and GD in verified SARS-CoV-2 positive individuals compared to age-matched controls. The data brings additional information on how to interpret previous studies based solely on subjective examination. Our results show that SARS-CoV-2 positive participants underestimate their subjective OD and GD compared to objective assessment. This underlines the importance of testing, and not just asking, if OD should be used for early detection of SARS-CoV-2. Among SARS-CoV-2 positive participants subjective OD/GD continued to be higher compared to negative controls at 30, 90 and 180 days follow up.

## Acknowledgments

We thank all the junior doctors who helped with inclusion. Furthermore, we thank The Acute Center, Zealand University Hospital Koege for providing facilities for inclusion of COVID-19 positive individuals. Furthermore, we thank the employees at the COVID test facilities helping promote the study to possible participants.

## Author Contributions

**Conceptualization:** Anne-Sophie Homøe, Elisabeth Arndal, Anders Koch, Grethe Badsberg Samuelsen, Xiaohui Chen Nielsen, Tobias Todsen, Preben Homøe.

**Data curation:** Mads Mose Jensen, Kasper Daugaard Larsen, Anne-Sophie Homøe, Anders Lykkemark Simonsen, Elisabeth Arndal, Anders Koch, Grethe Badsberg Samuelsen, Xiaohui Chen Nielsen, Tobias Todsen, Preben Homøe.

**Formal analysis:** Mads Mose Jensen, Kasper Daugaard Larsen, Preben Homøe.

**Funding acquisition:** Preben Homøe.

**Investigation:** Mads Mose Jensen, Grethe Badsberg Samuelsen, Tobias Todsen, Preben Homøe.

**Methodology:** Mads Mose Jensen, Elisabeth Arndal, Anders Koch, Grethe Badsberg Samuelsen, Xiaohui Chen Nielsen, Tobias Todsen, Preben Homøe.

**Project administration:** Mads Mose Jensen, Tobias Todsen, Preben Homøe.

**Resources:** Mads Mose Jensen, Anders Koch, Xiaohui Chen Nielsen, Preben Homøe.

**Software:** Mads Mose Jensen.

**Supervision:** Preben Homøe.

**Writing – original draft:** Mads Mose Jensen, Preben Homøe.

**Writing – review & editing:** Kasper Daugaard Larsen, Anne-Sophie Homøe, Anders Lykke-mark Simonsen, Elisabeth Arndal, Anders Koch, Grethe Badsberg Samuelsen, Xiaohui Chen Nielsen, Tobias Todsen, Preben Homøe.

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
