## [Decision Letter · Decision Letter 0]

11 Apr 2022

PONE-D-22-08137Subjective and objective olfactory and gustatory dysfunction among COVID-19 outpatients; Short- and long-term resultsPLOS ONE

Dear Dr. Jensen,

Thank you for submitting your manuscript to PLOS ONE. After careful consideration, we feel that it has merit but does not fully meet PLOS ONE’s publication criteria as it currently stands. Therefore, we invite you to submit a revised version of the manuscript that addresses the points raised during the review process.

ACADEMIC EDITOR: As appended below, the reviewers have raised major concerns/critiques (reviewer # 2 is against publication) and suggested further justification/work to consolidate the findings. Do go through the comments and amend the MS accordingly.

We look forward to receiving your revised manuscript.

Kind regards,

A. M. Abd El-Aty

Academic Editor

PLOS ONE

Journal Requirements:

Reviewers' comments:

Reviewer's Responses to Questions

**Comments to the Author**

1. Is the manuscript technically sound, and do the data support the conclusions?

Reviewer #1: Partly

Reviewer #2: Partly

Reviewer #3: Partly

2. Has the statistical analysis been performed appropriately and rigorously? 

Reviewer #1: Yes

Reviewer #2: Yes

Reviewer #3: Yes

3. Have the authors made all data underlying the findings in their manuscript fully available?

Reviewer #1: Yes

Reviewer #2: Yes

Reviewer #3: Yes

4. Is the manuscript presented in an intelligible fashion and written in standard English?

Reviewer #1: Yes

Reviewer #2: Yes

Reviewer #3: Yes

5. Review Comments to the Author

Reviewer #1: I have read this well-written manuscript with interest. The article prospectively analyzes a series of 58 COVID-19 patients by analyzing objectively (only at T0) and subjectively (with a follow-up of 6 months) the subjects recruited.

In my opinion a number of issues should be resolved before the article can be considered for publication:

1. In the first part of the introduction the references to the articles on the prevalence of OD and GD in general are dated and refer to articles from the first period of the pandemic (Sadeghat et al, Lechien et al.). Use references with broader case series (Lechien et al. 10.1007 / s00405-020-06548-w), with psychophysical studies (Vaira et al. 10.3390/pathogens10010062) and with revision with meta-analysis (Saniasiaya et al. 10.1002 / lary.29286)

2. Lines 55-56: "The pathogenesis for OD in COVID-19 is unknown, but it differs from other respiratory infections by being independent of nasal congestion" please add a reference.

3. Lines 56-61 Concerning the pathogenesis of olfactory disorders: ACE2 and TMPRSS receptors are mainly concentrated in supporting cells (Brann et al. 10.1126 / sciadv.abc5801; Lechien et al. 10.1007 / s12105-020-01212-5) but not in sensory neurons. The involvement of the bulb is suspected only on the basis of radiological case reports (Tan et al. 10.1002 / lary.30078) but not on the basis of histopathological reports (see the article cited by the authors of Khan et al.). The localized damage at the level of the olfactory epithelium is confirmed by all the histopathological reports currently present in the literature (kirschembaum et al. 10.1016 / S0140-6736 (20) 31525-7; Vaira et al. 10.1017 / S0022215120002455) and this dense with the clinical evidence of rapid recovery in most cases. Bulb atrophy could instead be secondary to a reduction in receptor impulses and not the primary cause of the dysfunction.

4. Both in the introduction and in the discussion the authors must acknowledge that robust studies already exist with 6 (Petrocelli et al. 10.1017/S002221512100116X; Hopkins et al. 10.4193/Rhin20.544, Boscolo-Rizzo et al. 10.1093/chemse/bjab006, Taziki Balajelini et al. 10.1017/S0022215121003935; Prem et al. 10.1007/s00405-021-07153-1, Niklassen et al. 10.1002/lary.29383, Otte et al. 10.1080/00016489.2021.1905178) and 12 month follow-up (Vaira et al. 10.1177/01945998211061511; Boscolo-Rizzo et al. 10.4193/Rhin21.249, Boscolo-Rizzo et al. 10.1007/s00405-021-06839-w). The results obtained by the authors need to be discussed in the light of the results of these other studies.

5. Line 66-67 This is incorrect as the study aims to understand the differences between subjective and objective tests during infection and to prospectively monitor recovery for up to 6 months with subjective methods only.

6. Lines 72-75 Have mechanisms to prevent inclusion bias been put in place? Could it be that subjects with olfactory or gustatory dysfunction were more motivated to participate in the study? If so, this must be recognized in the limitations of the study.

7. Were there any exclusion criteria? Were patients with prior known olfactory and gustatory dysfunctions or conditions predisposing to chemosensitive dysfunctions excluded?

8. How were the controls recruited? Did the controls have to be negative at the time of evaluation or have never contracted the infection before? How was negativity ascertained?

9. lines 113-114: hyposmia is a score between 6 and 8.

10. Lines 122-123 Questionnaire response rates at inclusion were 87.9% (51/58) for SARS-CoV-2 positives and 78.6% (44/56) for negative controls. What does it mean? If it means that the answers were 51 out of 58 why the 7 patients and the 12 controls who did not answer the questionnaire were not excluded from the analysis?

11. lines 145-146 if the previous presence of an olfactory or gustatory disturbance was an exclusion criterion for the enrollment of cases, why was it not also for the controls?

12. Has an analysis been carried out of the correlations of possible clinical and epidemiological risk factors with the persistence of chemosensitive disorders?

13. For the discussion please consider point 4.

14. One of the reasons for the different prevalence detected by the objective methods compared to the subjective ones may be that ODs are present in the general population with a frequency of about 20%. For this reason, subjects who do not know they have an OD report a normal sense of smell while a dysfunction is present on psychophysical tests.

15. The finding of a persistence of ODs in one third of cases at 6 months is a very important and worrying fact that deserves further discussion. Such a high prevalence of residual dysfunction, given the high prevalence of the infection in the population, means that we will have a large number of subjects with disabling long-term morbidity. This is even more serious if we think that:

a. chemosensory disturbances are also frequent in reinfections (Lechien et al. 10.1177 / 0145561320970105; Lechien et al. 10.1111 / joim.13259)

b. chemosensory disorders are also frequent in COVID-19 in vaccinated subjects (Vaira et al. 10.1002 / lary.29964)

c. chemosensory disturbances are also frequent with the omicron variant (10.1002 / alr. 22995)

d. persistent chemosensitive disorders are associated with a significant reduction in the quality of life of patients (Vaira et al. 10.3390 / life12020141, Saniasiaya et al. 10.1017 / S0022215121002279).

e. there are no shared guidelines on therapy.

16. In the limitations it must be added that the follow up was performed only with subjective tests which are not reliable in monitoring the recovery. Subjects who have had a great recovery may report a complete recovery while still being hyposmic (10.4193/Rhin21.249)

Reviewer #2: This study describes subjectively and objectively the effects of SARS-COV-2 infection to olfaction and gustation. This is really an interesting topic, however there are no findings to add new knowledge to what we already know. Also there is a major concern related to the exclusion criteria. According to my opinion patients with chronic sinus diseases or septal deviation that we know that permanently causes olfactory disorders (esp. hyposmia), as well as smokers that we know that smoking affects olfactory function should be excluded. Finally the number of patients (58) is small for safe results.

Reviewer #3: Congratulations to the authors for the study.

Below are some recommendations for the study.

Title

1) “Subjective and objective olfactory…” – The right term in not "objective olfactory test" but psychophysical tests. No objective assessment was performed in all fases of this study, please make this more clear.

Methods

Subjects

2) Line 73 – The sample included individuals aged between 18 and 80 years. However, it is known that the olfactory function, as in other sensory systems is impaired with aging. Therefore, it is very common for individuals after the age of 60 to present alterations in their sense of smell. The sample should have a lower age range, perhaps between 18 and 50 years, due to aging. In addition, these individuals did not undergo a smell assessment before being infected with COVID-19. What if, before the infection, they already had a slight or moderate change in the identification of odors, which they had not noticed? Therefore, it is important to review the age group.

3) Line 86 – I suggest removing “Objective tests” and putting “subjective tests”. See below.

4) Lines 95 and 96 – The authors state that the taste test - Burghart's Taste Stips and the olfactory test - Brief Smell 97 Identification Test are objective tests that were performed by an otolaryngologist. However, these tests are not objective, but subjective/psychophysical, as they depend on the individual's response. The only objective olfactory test is the event-related Olfactory Evoked Potential and the Electroolfactogram, which were not performed in the present sample.

5) Line 95 – How are these tests evaluated for normality and degrees of olfactory and gustatory loss? There is no description for the reader throughout the article.

6) Line 99 – No objective, gustatory or olfactory test was performed in this study. I recommend removing that phrase.

7) Line 107 – The authors state that a questionnaire was made to obtain demographic data and questions 1-6 of the SNOT-22 questionnaire were added. Why didn't you complete the SNOT-22 questionnaire, as it is validated and used internationally? It would be interesting and important to have carried out a complete assessment of the quality of life of each participant during the study period.

8) Line 109 – All participants answered this modified questionnaire four times during the study: at baseline, 30, 60, and 90 days? Is it possible to trust that these questionnaires were actually answered at these times since they were sent by e-mail? Were the psychophysical tests redone 30, 60, and 90 days after the first assessment? Or at least at the end of the study? This is not clear from the text.

Statistical methods

9) Line 112 – There are no objective methods for the assessment of smell and taste in this study.

10) Line 113 – The BSIT test is not objective, it is psychophysical.

Results

11) Lines 122 to 127 – I suggest putting this data in a table and drawing attention in writing only to the most important result.

12) Line 128 – The authors have already put in the title of the table what is being exposed here. I recommend removing and adding the caption.

13) In table 1 – it is necessary to place a legend below the table, for the acronyms PCR, OD, and GD and for other information that is necessary for the understanding of the table.

14) In table 1, in the “Reported symptoms and risk factors of OD/GD” part, add the symbol of (n) for the total number of participants and (%) to identify that the data are being presented in percentage.

15) In table 1, in the “Objective assessment” part, I suggest putting “ENT assessment” and also indicating that the data are presented as a percentage.

16) Line 134 – The acronym IQR is cited for the first time, but there is no full description of - Inter Quartile Range. This description will only occur on line 154.

Olfactory dysfunction

17) Lines 131 to 141 – There is no need to write down all the data that is already presented in the table. Comment only the most important result and tell the reader to analyze the data table.

18) Lines 142 and 143 – Table 2: This information would be better as a table title. I suggest improving the title and adding the legend below the table for the acronyms and other information that is necessary for the understanding of the table.

19) Lines 145 to 152 – the results described are in which table or graph? Please put the reference.

20) Lines 151 and 152: Please, make a table with the results of diagnostic tests for sensitivity, specificity, and positive predictive value.

21) Again, I recommend focusing only on the most important result and not writing out all the results that are already in the tables or graphs.

22) Line 157 (Figure 1) - I recommend improving the quality of the figure.

23) Line 161 - remove 'table 3' and leave only the explanation about the table and add the necessary captions.

Gustatory Dysfunction

24) Lines 167 to 175 – I recommend bringing the results of gustatory dysfunction together with the olfactory one so that the reader has table 2 just below for analysis. Do not describe the results already presented in the table in full. And don't forget to indicate the table.

25) Line 176 (Figure 2) - I recommend improving the quality of the figure.

Discussion

26) Line 181 – Replace “objective tests” with “psychophysical tests”.

27) Lines 181 to 185 – It is interesting to start the discussion by stating the main and most important result of the study. The description presented in this paragraph has already been done in the methodology.

28) Line 194 – “(…) suggesting that some individuals fail to recognize their OD.” This is important to point out, as many individuals cannot have a clear perception of how the sense of smell is.

29) In the discussion also review the description of what are subjective and objective tests, because as already mentioned above, all tests performed by the doctor were psychophysical and not objective. The previous questionnaire applied for the selection of participants is a self-report of olfaction.

30) Do not put so many results, as these are already in the tables. Discuss them only and focus only on the most important ones. Discussion is the crucial part to discuss the study data with what there is already literature.

Limitations

31) The authors report important limitations, mainly the bias in filling out the initial questionnaire.

Conclusion

32) Do not put statistical data in the conclusion. And do not summarize the study in this part.

33) The conclusion should be brief and answer your research question/objective. Do not put information that has already been discussed in the article or that is in the methodology.

34) Objective: “This study aimed to examine subjective and objective olfactory and gustatory function in non hospitalized individuals with acute COVID-19 up to 6 months after infection.” Answer in the conclusion: Was there an improvement in olfactory and gustatory function, after 6 months of the initial diagnosis of COVID?

35) Recommendation: send the article to be reviewed by a native speaker of English.

6. PLOS authors have the option to publish the peer review history of their article (what does this mean?). If published, this will include your full peer review and any attached files.

Reviewer #1: No

Reviewer #2: No

Reviewer #3: No

---

## [Author Response · Author response to Decision Letter 0]

7 Jul 2022

Reviewer comments Author reply Changes made

1. In the first part of the introduction the references to the articles on the prevalence of OD and GD in general are dated and refer to articles from the first period of the pandemic (Sadeghat et al, Lechien et al.). Use references with broader case series (Lechien et al. 10.1007 / s00405-020-06548-w), with psychophysical studies (Vaira et al. 10.3390/pathogens10010062) and with revision with meta-analysis (Saniasiaya et al. 10.1002 / lary.29286)

 Thank you for this excellent comment. We have added references for two separate meta-analysis. Line 50-52: Added: (1). However, the prevalence of OD ranges considerably from 5-80% of infected. (2,3). However, two separate meta-analysis show a similar prevalence at around 50% 

2. Lines 55-56: "The pathogenesis for OD in COVID-19 is unknown, but it differs from other respiratory infections by being independent of nasal congestion" please add a reference.

 Thank you for pointing out this unreferenced claim. Relevant references have been added. Inserted following references

Lee SH, Yeoh ZX, Sachlin IS, Gazali N, Soelar SA, Foo CY, m.fl. Self-reported symptom study of COVID-19 chemosensory dysfunction in Malaysia. Sci Rep. 8. februar 2022;12(1):2111. 

12. Salmon Ceron D, Bartier S, Hautefort C, Nguyen Y, Nevoux J, Hamel AL, m.fl. Self-reported loss of smell without nasal obstruction to identify COVID-19. The multicenter Coranosmia cohort study. J Infect. oktober 2020;81(4):614–20. 

3. Lines 56-61 Concerning the pathogenesis of olfactory disorders: ACE2 and TMPRSS receptors are mainly concentrated in supporting cells (Brann et al. 10.1126 / sciadv.abc5801; Lechien et al. 10.1007 / s12105-020-01212-5) but not in sensory neurons. The involvement of the bulb is suspected only on the basis of radiological case reports (Tan et al. 10.1002 / lary.30078) but not on the basis of histopathological reports (see the article cited by the authors of Khan et al.). The localized damage at the level of the olfactory epithelium is confirmed by all the histopathological reports currently present in the literature (kirschembaum et al. 10.1016 / S0140-6736 (20) 31525-7; Vaira et al. 10.1017 / S0022215120002455) and this dense with the clinical evidence of rapid recovery in most cases. Bulb atrophy could instead be secondary to a reduction in receptor impulses and not the primary cause of the dysfunction.

 Thank you for this interesting point. Indeed, you are correct that bulb atrophy seems secondary to olfactory dysfunction. We have changed the description of olfactory bulb involvement to avoid confusion. Line 62. “is related to” has been changes to “can possibly lead to” to better describe the involvement of the olfactory bulb. 

4. Both in the introduction and in the discussion the authors must acknowledge that robust studies already exist with 6 (Petrocelli et al. 10.1017/S002221512100116X; Hopkins et al. 10.4193/Rhin20.544, Boscolo-Rizzo et al. 10.1093/chemse/bjab006, Taziki Balajelini et al. 10.1017/S0022215121003935; Prem et al. 10.1007/s00405-021-07153-1, Niklassen et al. 10.1002/lary.29383, Otte et al. 10.1080/00016489.2021.1905178) and 

12 month follow-up (Vaira et al. 10.1177/01945998211061511; Boscolo-Rizzo et al. 10.4193/Rhin21.249, Boscolo-Rizzo et al. 10.1007/s00405-021-06839-w). The results obtained by the authors need to be discussed in the light of the results of these other studies.

 Thank you for these excellent references concerning duration of olfactory dysfunction. We have incorporated them and expanded the background section. Line 64-68: Added information regarding studies examining duration of dysfunction after 6 and 12 month. Furthermore, see questions 13 for changes to the discussion

5. Line 66-67 This is incorrect as the study aims to understand the differences between subjective and objective tests during infection and to prospectively monitor recovery for up to 6 months with subjective methods only.

 We appreciate this comment and acknowledge it can be interpreted as we test both subjective and objective for the follow up period. Line 72-74. Deleted “short and long term”

Added: and to monitor subjective recovery up to 6 months after infection. 

6. Lines 72-75 Have mechanisms to prevent inclusion bias been put in place? Could it be that subjects with olfactory or gustatory dysfunction were more motivated to participate in the study? If so, this must be recognized in the limitations of the study.

 Thank you for this very relevant comment. Indeed, we do have a risk of inclusion bias in this study, and the prevalence of olfactory and gustatory dysfunction would be better examined in a cross-sectional design. We fully acknowledge the risk of bias, and have added a line in the “limitation” section to clarify. Line 282: Added “which could possibly lead to inclusion bias” to clarify 

7. Were there any exclusion criteria? Were patients with prior known olfactory and gustatory dysfunctions or conditions predisposing to chemosensitive dysfunctions excluded?

 Thank you very much for this excellent comment. Participants were asked about prior olfactory or gustatory issues. However, we chose not to exclude participants with prior issues to better represent the background population, where chemosensitive disorders will also be prevalent. We recognize that this means we cannot conclude that reduced scores in psychophysical tests in SARS-CoV-2 positive individual with prior chemosensitive disorder is caused solely by SARS-CoV-2. However, by having a comparable control group we partly account for this issue and can still conclude that olfactory and gustatory dysfunction are more common amongst SARS-CoV-2 positive individuals. To better explain and highlight this we have added the few exclusion criteria and added the amount of participants who reported previous reduced sense of smell. 

Furthermore, we have made a further analysis, where we excluded all smokers, people with deviated septum and participants who answered they had a history of reduced smell or taste. This did not significantly change any results or conclusions. This information has been added in the discussion. Line 89: Added: Participants who could not fully comprehend the questionnaire and examination was excluded.

Line 278: In our study, participants with a self-reported history of OD/GD was not excluded. Likewise, smokers and individuals with deviated septum were not excluded. While we recognize this means that we cannot conclude SARS-CoV-2 is the cause of OD/GD for people with prior history or risk factors for OD/GD, we partly account for this by having a comparable control group. Furthermore, all statistical analysis were performed where all individuals with prior OD/GD, septal deviation, smoking history was excluded. This did not significantly change the results or conclusions and it was decided to keep them in the final analysis. 

Table 1: Added column regarding previous olfactory/gustatory dysfunction. 

8. How were the controls recruited? Did the controls have to be negative at the time of evaluation or have never contracted the infection before? How was negativity ascertained?

 Controls were also included through PCR-test facilities in the two regions. All participants had three further PCR test from different anatomical localizations at inclusion to assert positivity/negativity. 

Unfortunately, we did not have any information regarding previous infection with SARS-CoV-2. Line 101: added “on both cases and controls” to clarify.

Line 103: Information regarding previous infections with SARS-CoV-2 was not available. 

9. lines 113-114: hyposmia is a score between 6 and 8.

 Thank you for the comment. The range has been corrected. Line 108-110: The degree of psychophysical OD (BSIT test) was categorized into either normosmia (BSIT score ≥9/12), hyposmia (≤8 - ≥6) or anosmia (≤5/12).

Per request of reviewer 2, the section has been moved to a different subheading.

10. Lines 122-123 Questionnaire response rates at inclusion were 87.9% (51/58) for SARS-CoV-2 positives and 78.6% (44/56) for negative controls. What does it mean? If it means that the answers were 51 out of 58 why the 7 patients and the 12 controls who did not answer the questionnaire were not excluded from the analysis?

 Thank you for making this point. As you propose, the answer rate means that 7 patients and 12 controls did not complete the online questionnaire. However, all participants completed the objective assessment and psychophysical tests. We therefore report results for all patients and follow-up results for the exact number who answered the questionnaire (table 4) No changes made

11. lines 145-146 if the previous presence of an olfactory or gustatory disturbance was an exclusion criterion for the enrollment of cases, why was it not also for the controls?

 Thank you for your comment. Please see answer 7 where a detailed explanation is given regarding exclusion. No changes made.

12. Has an analysis been carried out of the correlations of possible clinical and epidemiological risk factors with the persistence of chemosensitive disorders?

 Thank you for this excellent comment. It would be very interesting to examine risk factors for persistent olfactory/gustatory dysfunction. However, as the number of participants who completed the follow-up questionnaire was quite low, we concluded that subgroup analysis would have a high risk of bias and not give any valid results. 

13. For the discussion please consider point 4.

 Again, we thank you for these great references. We have added them in the discussion and expanded the discussion. Line: 252-253: Added references and “while, up to a quarter of previously infected report persistent olfactory dysfunction.”

14. One of the reasons for the different prevalence detected by the objective methods compared to the subjective ones may be that ODs are present in the general population with a frequency of about 20%. For this reason, subjects who do not know they have an OD report a normal sense of smell while a dysfunction is present on psychophysical tests.

 Thank you for this excellent point. We completely agree that unrecognized olfactory dysfunction is present in the background population as well. We have added a sentence in the discussion related to this point to clarify. Line. 232-235: Added: “This discrepancy is also known from the general population, where psychophysical tests reveal up to 20% of the background population suffer from OD, but only around 10% report subjective OD”

15. The finding of a persistence of ODs in one third of cases at 6 months is a very important and worrying fact that deserves further discussion. Such a high prevalence of residual dysfunction, given the high prevalence of the infection in the population, means that we will have a large number of subjects with disabling long-term morbidity. This is even more serious if we think that:

a. chemosensory disturbances are also frequent in reinfections (Lechien et al. 10.1177 / 0145561320970105; Lechien et al. 10.1111 / joim.13259)

b. chemosensory disorders are also frequent in COVID-19 in vaccinated subjects (Vaira et al. 10.1002 / lary.29964)

c. chemosensory disturbances are also frequent with the omicron variant (10.1002 / alr. 22995)

d. persistent chemosensitive disorders are associated with a significant reduction in the quality of life of patients (Vaira et al. 10.3390 / life12020141, Saniasiaya et al. 10.1017 / S0022215121002279).

e. there are no shared guidelines on therapy.

 Thank you for these interesting points. We completely agree that the number of possible individuals is overwhelming, and recognize we have not fully discussed this. We have added a paragraph in the discussion to highlight this public health concern. Line 257-262: Added: “While there is still some uncertainty how the long term recovery differs between different strains of SARS-CoV-2 and how vaccinations influences the long term recovery (33), the high rates of persistent olfactory dysfunction is very concerning. A large proportion of the world’s populations is likely previously infected with SARS-CoV-2, and the total number of individuals at risk of persistent chemosensetive dysfunction is enormous. These individuals face a reduced quality of life, and this is of significant public concern. Consensus on treatment guidelines are lacking”

16. In the limitations it must be added that the follow up was performed only with subjective tests which are not reliable in monitoring the recovery. Subjects who have had a great recovery may report a complete recovery while still being hyposmic (10.4193/Rhin21.249) We highly appreciate the reviewer has pointed out this weakness in our study. We agree that this design is not optimal to monitor recovery. We have added a paragraph under the limitation section. Line 297-299: Added: “Our follow up was limited to subjective answers from questionnaires. As already described, there is a discrepancy between subjective and psychophysical testing, and the long term recovery results should be interpreted with care.”

Reviewer #2: 

This study describes subjectively and objectively the effects of SARS-COV-2 infection to olfaction and gustation. This is really an interesting topic, however there are no findings to add new knowledge to what we already know. Also there is a major concern related to the exclusion criteria. According to my opinion patients with chronic sinus diseases or septal deviation that we know that permanently causes olfactory disorders (esp. hyposmia), as well as smokers that we know that smoking affects olfactory function should be excluded. Finally the number of patients (58) is small for safe results. This case-control study aimed to describe OD in a population-based setting in persons showing up in the public test facilities and who did not need hospital care. We believe this approach adds new information and that our study also adds to some other newly reported findings. We chose that previous illness and relevant co-morbidities were registered for all participants instead of excluding individuals with septal deviation etc. We argue the case-control design allows for this approach but also performed separate analysis where individuals with previous chemosensitive disorders, septal deviation which did not significantly change the results. A paragraph have been added in the discussion. No changes in the manuscript

Line 278: In our study, participants with a self-reported history of OD/GD was not excluded. Likewise, smokers and individuals with deviated septum were not excluded. While we recognize this means that we cannot conclude SARS-CoV-2 is the cause of OD/GD for people with prior history or risk factors for OD/GD, we partly account for this by having a comparable control group. Furthermore, all statistical analysis were performed where all individuals with prior OD/GD, septal deviation, smoking history was excluded. This did not significantly change the results or conclusions and it was decided to keep them in the final analysis. 

Reviewer #3: 

Congratulations to the authors for the study.

Below are some recommendations for the study.

Title

 Thank you for the kind remarks 

1) “Subjective and objective olfactory…” – The right term in not "objective olfactory test" but psychophysical tests. No objective assessment was performed in all fases of this study, please make this more clear.

 We appreciate this comment very much, as it does indeed suggest we performed true objective testing. We agree that psychophysical test is a more accurate term and have corrected the term throughout the article. Title: Changed “objective” to “psychophysical”

Methods

Subjects

2) Line 73 – The sample included individuals aged between 18 and 80 years. However, it is known that the olfactory function, as in other sensory systems is impaired with aging. Therefore, it is very common for individuals after the age of 60 to present alterations in their sense of smell. The sample should have a lower age range, perhaps between 18 and 50 years, due to aging. In addition, these individuals did not undergo a smell assessment before being infected with COVID-19. What if, before the infection, they already had a slight or moderate change in the identification of odors, which they had not noticed? Therefore, it is important to review the age group.

 Thank you for this relevant comment. We have by the case-control design with age-matching of positive and negative persons attempted to overcome the mentioned problems. Changes in manuscript line 88 “Age-matched” is inserted and lines 89-90 

3) Line 86 – I suggest removing “Objective tests” and putting “subjective tests”. See below.

 See answer 1. Line 70 Changes “objective” to “psychophysical”

4) Lines 95 and 96 – The authors state that the taste test - Burghart's Taste Stips and the olfactory test - Brief Smell 97 Identification Test are objective tests that were performed by an otolaryngologist. However, these tests are not objective, but subjective/psychophysical, as they depend on the individual's response. The only objective olfactory test is the event-related Olfactory Evoked Potential and the Electroolfactogram, which were not performed in the present sample.

 See answer 1 Line 94 Changes “objective” to “psychophysical”

5) Line 95 – How are these tests evaluated for normality and degrees of olfactory and gustatory loss? There is no description for the reader throughout the article.

 We thank you for pointing this out. Initially, we state the scores under “statistical methods”, but see that it is better suited to follow the description of the different tests. We have moved the description and added a short description regarding Burgharts taste test. Line 106: Failure to recognize all 4 flavors was considered as gustatory dysfunction.

Line 106-108: Moved the following section “The degree of psychophysical OD (BSIT test) was categorized into either normosmia (BSIT score ≥9/12), hyposmia (≤8 - ≥6) or anosmia (≤5/12).”

6) Line 99 – No objective, gustatory or olfactory test was performed in this study. I recommend removing that phrase.

 We agree. See answer 1 

7) Line 107 – The authors state that a questionnaire was made to obtain demographic data and questions 1-6 of the SNOT-22 questionnaire were added. Why didn't you complete the SNOT-22 questionnaire, as it is validated and used internationally? It would be interesting and important to have carried out a complete assessment of the quality of life of each participant during the study period.

 Thank you very much for this point. It would have been very interesting to evaluate quality of life throughout the study period. However, as we also included several other questions, we feared the questionnaires would be too comprehensive and we feared this would deter participants from completing all follow-up questionnaires. As such, we chose only to include the nasal symptoms related questions from SNOT-22 Line 119: Added “The nasal symptoms related”

8) Line 109 – All participants answered this modified questionnaire four times during the study: at baseline, 30, 60, and 90 days? Is it possible to trust that these questionnaires were actually answered at these times since they were sent by e-mail? Were the psychophysical tests redone 30, 60, and 90 days after the first assessment? Or at least at the end of the study? This is not clear from the text. We thank you very much for this comment. The questionnaires itself was hosted online through REDCap, and the email only serves as an invitation with a link. Therefore, we have date and time for all answers. The follow-up was only subjective answers. We have highlighted this in the method section to clarify. Line 73-74: and to monitor subjective recovery up to 6 months after infection.

Statistical methods

9) Line 112 – There are no objective methods for the assessment of smell and taste in this study. We agree. Please see answer 1 No changes in the manuscript

10) Line 113 – The BSIT test is not objective, it is psychophysical. See answer 1 No changes in the manuscript

Results

11) Lines 122 to 127 – I suggest putting this data in a table and drawing attention in writing only to the most important result. We thoroughly appreciate the review of the results section. Overall, we agree that there are many results written in the text and have followed your suggestion to only bring forward the most interesting results. Line 133: Deleted section and replaced with “can be seen in Table 3”

12) Line 128 – The authors have already put in the title of the table what is being exposed here. I recommend removing and adding the caption.

 We appreciate the comments on table 1 from the reviewer. We agree that the title is quite long. We have reduced the length and instead put the description in the caption. Table 1: Corrected title. Added “and risk factors for reduced sense of smell and taste for SARS-CoV-2 PCR positive cases and negative controls.” In caption

13) In table 1 – it is necessary to place a legend below the table, for the acronyms PCR, OD, and GD and for other information that is necessary for the understanding of the table.

 Thank you for pointing out these acronyms, which are not explained in the table. We have added a description in the caption Table 1. Added: “OD: Olfactory dysfunction, GD: Gustatory dysfunction, ENT: Ear Nose Throat” in caption.

14) In table 1, in the “Reported symptoms and risk factors of OD/GD” part, add the symbol of (n) for the total number of participants and (%) to identify that the data are being presented in percentage.

 Thank you very much for point out the faulty table. We have followed your advice to clarify. Table 1: Added n/N (%) throughout

15) In table 1, in the “Objective assessment” part, I suggest putting “ENT assessment” and also indicating that the data are presented as a percentage.

 Thank you for this comment. We agree that the term objective can be ambiguous and have replaced it with “ENT”. Table 1: Deleted “Objective”. Added “ENT”

16) Line 134 – The acronym IQR is cited for the first time, but there is no full description of - Inter Quartile Range. This description will only occur on line 154. Thank you for the comment. You are correct that the description is written to late. Due to reduction in the result section, the description is now mentioned at the first use. As suggested, results are now mainly reported in tables and IQR is not mentioned in text. In tables, IQR is written out.

Olfactory dysfunction

17) Lines 131 to 141 – There is no need to write down all the data that is already presented in the table. Comment only the most important result and tell the reader to analyze the data table. We thank the reviewer very much for the comments regarding the result section. We agree the section is extensive and have shortened the whole section. Line 151-156 deleted.

18) Lines 142 and 143 – Table 2: This information would be better as a table title. I suggest improving the title and adding the legend below the table for the acronyms and other information that is necessary for the understanding of the table.

 We thank you for pointing out the poor title of the table. We have corrected both title and caption of the table. Table 2: Changes title and caption

19) Lines 145 to 152 – the results described are in which table or graph? Please put the reference.

 Thank you for pointing out the missing reference. We have added the required information. Line 1181: Added: Table 4

20) Lines 151 and 152: Please, make a table with the results of diagnostic tests for sensitivity, specificity, and positive predictive value.

 We thank you for suggesting making another table to fit these results. We agree that it helps with the understanding. Inserted result into table 3

21) Again, I recommend focusing only on the most important result and not writing out all the results that are already in the tables or graphs.

 Again, we thank you for pointing out the extensive result section. As already described, we have shortened the section throughout. Line174-179: Deleted results and referred to table instead. 

Line 152-156: deleted results and referred to table 2.

Line 180-181 deleted results and referred to table 2. 

22) Line 157 (Figure 1) - I recommend improving the quality of the figure.

 Thank you for pointing out the quality of the figure is poor. We suspect the image quality reduced during conversion to the required format. We will contact the journal to resubmit figures with better quality. No changes made.

23) Line 161 - remove 'table 3' and leave only the explanation about the table and add the necessary captions.

Gustatory Dysfunction

 Thank you for this comment regarding table 3, now table 4. We have edited the legend of the table. Furthermore, we discovered some inconsistencies in the subheadings and have corrected them. Table 4: Corrected subheading and legend.

24) Lines 167 to 175 – I recommend bringing the results of gustatory dysfunction together with the olfactory one so that the reader has table 2 just below for analysis. Do not describe the results already presented in the table in full. And don't forget to indicate the table.

 We thank you for this relevant comment, and agree the table separates the section. However, the journal guidelines suggest inserting the table after the first mention. However, I suspect the table placement will differ in the final print. 

We have shortened the result section and instead refer to the relevant tables. Line 204-207: Deleted results. Added “and negative controls can be seen in table 4 and figure 2”

25) Line 176 (Figure 2) - I recommend improving the quality of the figure.

 See answer 22. No changes made

Discussion 

26) Line 181 – Replace “objective tests” with “psychophysical tests”.

 See answer 1 Deleted due to answer 27.

27) Lines 181 to 185 – It is interesting to start the discussion by stating the main and most important result of the study. The description presented in this paragraph has already been done in the methodology.

 Thank you for this excellent point. We agree that the discussion is best started off with key results, and the current start has already been mentioned. We have deleted the section. Line 213-217: deleted

28) Line 194 – “(…) suggesting that some individuals fail to recognize their OD.” This is important to point out, as many individuals cannot have a clear perception of how the sense of smell is. We are highly appreciative of this comment. We agree that some may have poor perception of their own sense of smell. We have already added a sentence to accommodate a comment from reviewer 1 and think this elaboration also describes you comment. Line 233-235: Added “This discrepancy is also known from the general population, where psychophysical tests reveal up to 20% of the background population suffer from Olfactory dysfunction, but only around 10% report subjective olfactory dysfunction, suggesting many individuals do not have a clear perception of their sense of smell”

Limitations 

29) In the discussion also review the description of what are subjective and objective tests, because as already mentioned above, all tests performed by the doctor were psychophysical and not objective. The previous questionnaire applied for the selection of participants is a self-report of olfaction.

 We highly appreciate this comment regarding subjective and objective testing, and agree it is important to discuss the limitation. We have added a section under “limitations”. Line 290-296: added “It is important to recognize that BSIT and other psychophysical tests are not true objective test of an individual’s olfaction. However, true objective test for olfactory dysfunctions, such as olfactory evoked potential, is not ideal for large group of patients and especially not for SARS-CoV-2 infected individuals. Therefore psychophysical testing is often used as a tool for quantifying sense of smell. This quantification is very useful, and often uncovers olfactory dysfunctions not recognized by the patient. However, when interpreting the results one should recognize that answers depend on the individual performing the test.”

30) Do not put so many results, as these are already in the tables. Discuss them only and focus only on the most important ones. Discussion is the crucial part to discuss the study data with what there is already literature.

 Thank you again for point out our result and discussion section often repeat results. We completely agree that it is better to refer to tables instead. We have shortened the discussion section accordingly. Line 240: Deleted “78.5% and 87%.”

Line 248-249: Deleted results

31) The authors report important limitations, mainly the bias in filling out the initial questionnaire. Indeed, this study does have limitation, and as you describe, inclusion bias is an issue. Per request of reviewer 1, this point was made clearer in the limitation section. Line 301: added “which could possibly lead to inclusion bias”

 During the revision, we found an error in the result section and corrected the number. 200: Error in number. Corrected to 56.9% (33/58)

---

## [Decision Letter · Decision Letter 1]

20 Sep 2022

Subjective and psychophysical olfactory and gustatory dysfunction among COVID-19 outpatients; Short- and long-term results

PONE-D-22-08137R1

Dear Dr. Jensen,

We’re pleased to inform you that your manuscript has been judged scientifically suitable for publication and will be formally accepted for publication once it meets all outstanding technical requirements.

Kind regards,

A. M. Abd El-Aty

Academic Editor

PLOS ONE

Additional Editor Comments (optional):

Reviewers' comments:

Reviewer's Responses to Questions

**Comments to the Author**

1. If the authors have adequately addressed your comments raised in a previous round of review and you feel that this manuscript is now acceptable for publication, you may indicate that here to bypass the “Comments to the Author” section, enter your conflict of interest statement in the “Confidential to Editor” section, and submit your "Accept" recommendation.

Reviewer #1: All comments have been addressed

2. Is the manuscript technically sound, and do the data support the conclusions?

Reviewer #1: Yes

3. Has the statistical analysis been performed appropriately and rigorously? 

Reviewer #1: Yes

4. Have the authors made all data underlying the findings in their manuscript fully available?

Reviewer #1: Yes

5. Is the manuscript presented in an intelligible fashion and written in standard English?

Reviewer #1: Yes

6. Review Comments to the Author

Reviewer #1: Authors satisfactorily responded to all my observations and comments. In my opinion, the article can be published.

7. PLOS authors have the option to publish the peer review history of their article (what does this mean?). If published, this will include your full peer review and any attached files.

Reviewer #1: No

---

## [Editor Report · Acceptance letter]

22 Sep 2022

PONE-D-22-08137R1 

Subjective and psychophysical olfactory and gustatory dysfunction among COVID-19 outpatients; Short- and long-term results. 

Dear Dr. Jensen:

I'm pleased to inform you that your manuscript has been deemed suitable for publication in PLOS ONE. Congratulations! Your manuscript is now with our production department. 

Kind regards, 

on behalf of

Prof. A. M. Abd El-Aty 

Academic Editor

PLOS ONE